# Text Classification Model Enhanced by Unlabeled Data for LaTeX Formula

**Hua Cheng *** , **Renjie Yu** , **Yixin Tang, Yiquan Fang and Tao Cheng**

School of Information Science and Engineering, East China University of Science and Technology, 130 Meilong Road, Shanghai 200237, China; y45190204@mail.ecust.edu.cn (R.Y.); y30200952@mail.ecust.edu.cn (Y.T.); fyq@ecust.edu.cn (Y.F.); y80210026@mail.ecust.edu.cn (T.C.)
* Correspondence: hcheng@ecust.edu.cn

**Abstract:** Generic language models pretrained on large unspecific domains are currently the foundation of NLP. Labeled data are limited in most model training due to the cost of manual annotation, especially in domains including massive Proper Nouns such as mathematics and biology, where it affects the accuracy and robustness of model prediction. However, directly applying a generic language model on a specific domain does not work well. This paper introduces a BERT-based text classification model enhanced by unlabeled data (UL-BERT) in the LaTeX formula domain. A two-stage Pretraining model based on BERT(TP-BERT) is pretrained by unlabeled data in the LaTeX formula domain. A double-prediction pseudo-labeling (DPP) method is introduced to obtain high confidence pseudo-labels for unlabeled data by self-training. Moreover, a multi-rounds teacher–student model training approach is proposed for UL-BERT model training with few labeled data and more unlabeled data with pseudo-labels. Experiments on the classification of the LaTex formula domain show that the classification accuracies have been significantly improved by UL-BERT where the F1 score has been mostly enhanced by 2.76%, and lower resources are needed in model training. It is concluded that our method may be applicable to other specific domains with enormous unlabeled data and limited labelled data.

**Keywords:** unlabeled data; self-training; pretraining; BERT; LaTeX formula

## 1. Introduction

In most domains, it is a common problem where labeled data are precious and scarce due to costly human resources, but unlabeled data are easily sampled. Particularly, in professional fields such as LaTeX or biomedicine, accurate labels are limited, costly and easily misclassified since they need people with field knowledge. By leveraging massive unlabeled data and fewer resources, an effective classifier can be proposed and optimized for better labelling. Mining information from unlabeled data to improve the performance of classification models is needed. A large amount of unlabeled data are utilized to improve the model's robustness by self-training without much extra complexity, which has been successfully applied to image recognition [1,2] and speech recognition [3–5]. The self-training method labels unlabeled data with a teacher model and uses the labeled data and unlabeled data to jointly train a student model. Reference [2] proposed that the strategy of Noisy Student Training was helpful for improving the generalization capabilities and robustness. Reference [4] proposed the normalized filtering scores that filter out low-confident utterance-transcript pairs generated by the teacher to mitigate the noise introduced by the teacher model.

On the other hand, in the text classification task of specific domains, such as biology, information security, and mathematics, where Proper Nouns are widely used, pretraining from unlabeled data is more critical, which can make the representation of terms more accurate in NLP tasks. The representation learning of a specific domain can be enhanced by the pretraining [6] of EMLO [7] or BERT [8]. BioBERT [9] was pretrained in the biomedical

field, which can have a greater performance in the corresponding biomedical text mining task. MathBERT [10] was proposed for mathematical formulas, and a new pretraining task was designed to improve the model's ability. Reference [11] also showed the effectiveness of the pretraining of a specific domain in computer vision, where a deep learning model was trained on a large unlabeled medical image dataset. Better results can be obtained on a small number of labeled medical images.

Recently, semi-supervised learning [12] has become a new research direction that has attracted much attention in deep learning. In SimCLRv2 [13], unsupervised pretraining of a ResNet architecture and self-training via unlabeled examples for refining and transferring the task-specific knowledge are proposed for image recognition. Self-training and pretraining are complementary and effective approaches to improve speech recognition using unlabeled data [14].

The widely used semi-supervised methods are mainly based on pseudo-labels [15,16], which are predicted by models (such as decision tree, generative model, BERT) on unlabeled data. The unlabeled data with pseudo-labels and the original data form a new dataset, which is used to retrain the original model. Self-training is a specific type of pseudo-label-based method, which includes a teacher model for pseudo-labeling, and a student model on the new dataset. In the self-training process, the teacher model is updated by the student model in each training round to further improve the model's performance. It is necessary to explore more values of unlabeled data in specific text domains on classification tasks by means of pretraining and self-training.

This paper proposes a text classification model (UL-BERT) in the LaTeX formula domain that takes advantage of unlabeled data and requires fewer labeled data. The word vector space is transferred from the general space of BERT to a particular space pretrained by domain pretrained stage and task pretrained stage on unlabeled data. A new dataset on unlabeled data with pseudo-labels is produced, and fed to a multi-round model training process to improve the performance of the text classification model. In this paper, the LaTex formula's data are chosen as the study case to investigate the role of unlabeled data in the text classification task.

In summary, the contribution of this work is as follows:

- We propose a two-stage pretraining model based on BERT(TP-BERT) for the domain specific unlabeled data. It ensures the word vector's space is more effective for the representation of the data in the LaTeX formula domain;
- We introduce the double-prediction pseudo-labeling (DPP) method to obtain more accurate pseudo-labels. The pseudo-labels are selected, which are predicted by the same labels from the origin and its augmented unlabeled text and have high predicting probabilities. It improves the confidence of the pseudo-label and reduces the effects from the label noise;
- We propose a multi-rounds teacher–student training approach to train the classification model, where pseudo-labels are iteratively predicted by the new student model. It shows that our training approach can achieve a superior performance and low resources are needed.

## 2. Pretraining Based on Unlabeled Data

The pretrained models, such as BERT, are based on a general large-scale corpus such as BooksCorpus and English Wikipedia. We pretrained a two-stage Pretraining BERT model (TP-BERT) on unlabeled text data, including a domain pretraining stage and then a task pretraining stage. By adding the domain data to the pretraining dataset, the pretrained model can learn more targeted information from the domain data and the representation is more accurate in finetune.

The domain pretraining stage makes the pretrained model adapt more to the domain data, such as in BioBERT [9], a linguistic representation model for biomedicine. The task pretraining stage pretrains on the task dataset, a narrowly-defined subset of the domain itself or data relevant to the task, which may be helpful for a specific task pretrained model.



Figure 1 shows the detail of the TP-BERT model of LaTeX formulas. BERT is trained with a masked language modeling objective to obtain TP-BERT based on domain data (LaTeX formulas in thesis) and task data (LaTeX formulas in mathematics thesis).

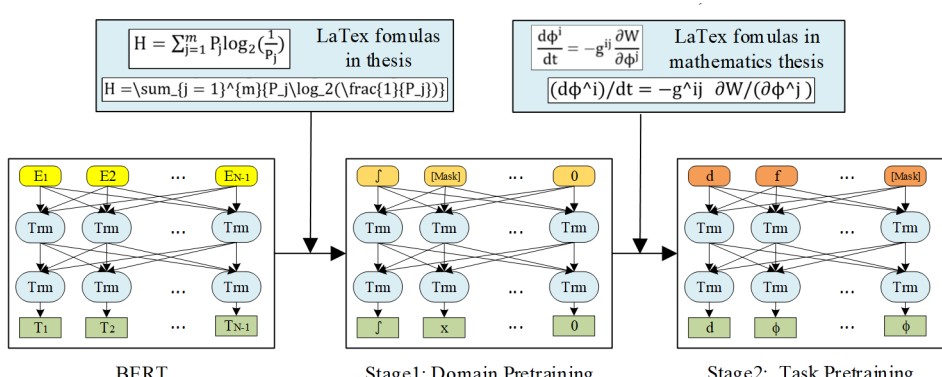

**Figure 1.** Two-stages Pretraining BERT model (TP-BERT).

Figure 2 is the corresponding word vector spaces of BERT and TP-BERT on LaTeX mathematical formulas shown by UMAP [17]. It demonstrates that the word vector of TP-BERT has changed after domain pretraining and task pretraining, where the red dots move down and left from the purple ones.

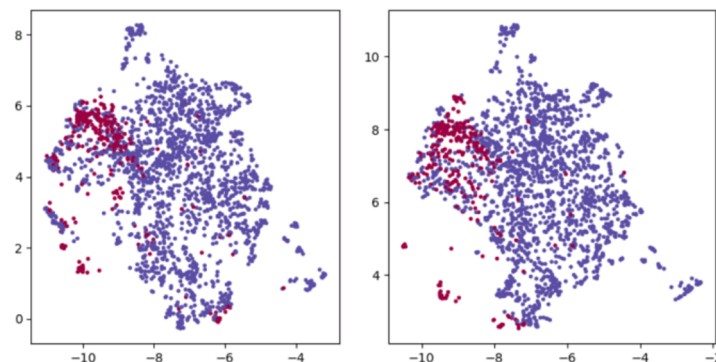

**Figure 2.** Word vectors spaces of BERT and TP-BERT. Red: word vectors of LaTeX mathematical formulas; Purple: other word vectors.

### 3. Unlabeled Data Enhanced Text Classification

*3.1. Double-Prediction Pseudo-Labeling Method*

Besides the role in pretraining, a new pseudo-label dataset can be produced based on unlabeled data by self-training, which is a helpful and effective in-model fine-tuning. The self-training method has been proposed for image recognition [2], where it assigns pseudo-labels for unlabeled images by way of a noisy and inaccurate model which is only trained on a few labeled data. This noisy model is improved and retrained on both labeled and unlabeled images.

A revised self-training method on unlabeled text data is proposed to generate a high confidence pseudo-label from an input text and its variant. Considering the immaturity of the initial model trained with few labeled data, we improve the robustness of pseudo-label prediction, where the Double-Prediction Pseudo-labeling (DPP) method is proposed to obtain high confidence Pseudo-labels predicted by the last trained model.

The original training data are defined as $x_i$ with label $y_i$, unlabeled data $\tilde{x}_i$. The variants, the augmented unlabeled data $\tilde{x}_i'$, are obtained by adding auxiliary sentences to $\tilde{x}_n$. Auxiliary sentences are meaningless texts that do not affect classification, such as 'Without using a calculator, ...' to $\tilde{x}_n$. The DPP method in self-training is as follows:

1. The UL-BERT model is fine-tuned based on TP-BERT by the original training dataset $(x_i, y_i)$. The cross-entropy loss function $L_t$ is defined by:

$$L_t = -\frac{1}{m} \sum_i^m \sum_j^l y_i^j log p_i^j,$$ (1)

   where $m$ is the batch size and $l$ is the number of and classes, $y$ is the ground truth, and $p$ is the predicted label.

2. Labels $\widetilde{y}_i$ and $\widetilde{y}_i'$ are predicted based on an unlabeled dataset $\widetilde{x}_i$ and $\widetilde{x}_i'$ by UL-BERT.

$$\widetilde{y}_i = f(\widetilde{x}_i, \theta^t)$$ (2)

$$\widetilde{y}_i' = f(\widetilde{x}_i', \theta^t).$$ (3)

3. For data with the same predicted labels whose prediction probabilities are both above a specific threshold $t$, label $\widetilde{y}_i^p$ is taken as its pseudo-label. The higher prediction probability has better class confidence.

$$\widetilde{y}_i^p = \begin{cases} \widetilde{y}_i', & \widetilde{y}_i = \widetilde{y}_i' and \widetilde{p}_i >= t and \widetilde{p}_i' >= t \\ null, & other, \end{cases}$$ (4)

   where $\widetilde{p}_i$ is the predicted probability, and $t$ is the threshold.

4. The new dataset $\left(\widetilde{x}_i^p, \widetilde{y}_i^p\right)$ with pseudo-labels is obtained and combined with the original label dataset to train the classification model. In general, the problem of few labeled datasets can be solved by retraining UL-BERT with the new pseudo-labeled dataset from unlabeled data self-training, which improves the robustness, generalization and accuracy of the classification model.

### 3.2. Multi-Rounds Teacher–Student Training Approach

The classification model should be trained for multi-rounds since the pseudo-labels from the single-round self-training are not accurate enough. The multi-rounds teacher–student training approach is proposed, where the Teacher model generates the pseudo-labels to retrain the Student model, and the Student model updates the Teacher model on the next training round.

The role of the Teacher model is to help and supervise the training of a better Student model. It can reduce the prediction instability caused by an immature model and improve its classification accuracy.

Figure 3 shows the training approach, and the training process is as follows:

1. TP-BERT is pretrained by the domain and task unlabeled data $\{\widetilde{x}_1, \widetilde{x}_2, \ldots \widetilde{x}_i, \ldots, \widetilde{x}_n\}$;
2. For the first time, UL-BERT (Teacher) is fine-tuned based on TP-BERT by the original labeled data $(x_1, y_1), (x_2, y_2), \ldots, (x_n, y_n)$;
3. By way of the DPP method, a high confidence label will be selected as an available pseudo-label $\widetilde{y}_i^p$, which can be expressed as:

$$\widetilde{y}_i^p = DPP(\widetilde{x_i^p});$$ (5)

4. The UL-BERT model (Student) is fine-tuned on TP-BERT using the new dataset $\left(\widetilde{x}_i^p, \widetilde{y}_i^p\right)$ and the original training data $(x_i, y_i)$. The cross-entropy loss function $L_s$ from both data is given by:

$$L_s = -\frac{1}{m} \sum_{j=1}^m \sum_{i=1}^l y_j^i log p_j^i - \frac{1}{k} \sum_{j=1}^n \sum_{i=1}^l \widetilde{y}_j^i log \widetilde{p}_j^i,$$ (6)

where *m* and *k* are the batch size of the label data and pseudo-label data, which are selected from $(\widetilde{x_n}, \widetilde{y_n})$; $\widetilde{y}$ is the pseudo-label and $\widetilde{p}$ is the pseudo-labeled data predicted label;

5.   If the F1 score of the test dataset (by UL-BERT Student model) is better than the one (by UL-BERT Teacher model), the Teacher model will be updated by the Student model;

6.   Repeat steps 3–5 until the F1 score of the test dataset of the Student model no longer increases.

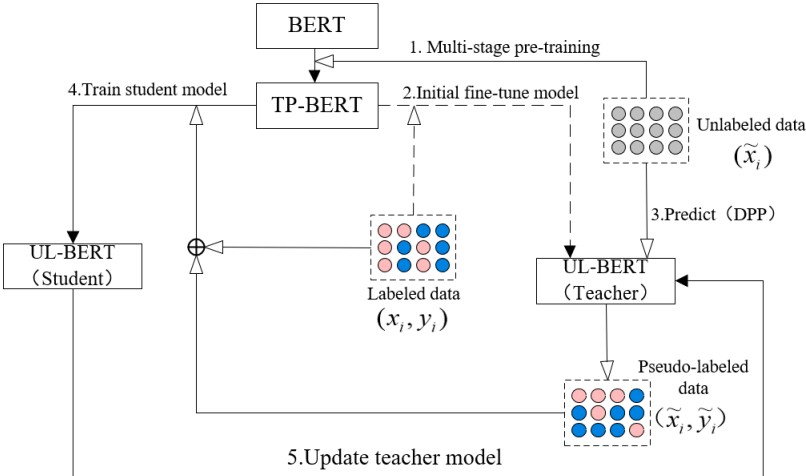

**Figure 3.** Teacher-Student (T–S) multi-rounds training of classification model (UL-BERT).

## 4. Experiments and Discussion

### 4.1. Datasets

Due to the Proper Nouns in LaTeX commands, LaTeX formulas are quite different from the large-scale corpus pretrained in BERT. Therefore, LaTeX formulas are selected as the study case for the domain-specific text classification.

The Kanervisto dataset [18], as the 'LaTeX formulas' unlabeled dataset, contains 104,564 LaTeX mathematical formulas from published papers. The 'LaTeX formulas of Mathematics' labeled data from other papers are classified in Table 1, which has 4573 Mathematics formulas. The ratio of the original training, validation, and test datasets is 8:1:1 in Table 2.

Derivative, Integral, Series, and Matrix are the main basic formulas in Mathematics. Since most formulas in the labeled dataset are composites of basic formulas, the annotation rules are defined as follows:

- Inclusion relationship $X(Y)$
  $Y$ is calculated as an element of $X$, the composite formula $X(Y)$ is annotated in the same class as that of $X$. $X$ and $Y$ can be any basic formula or composite formula $X(Y)$. For example, $\int x'$ is annotated as the 'Integral' class;

- Combination relationship $\odot(X, Y)$
  The composite formula is annotated as the same class when X and Y are the same class. Otherwise, it should be annotated as the 'Others' class. $\odot$ represents arithmetic operators, such as addition, subtraction, multiplication, and relational operators such as greater than, less than.

- Other relationship
  The non-basic formulas will be annotated as the 'Others' class. As the labeled data demonstrated in Table 1, the formulas in our dataset match with the labeling rule. For example, the formula of Average mutual information in the 'Communication Principle' $I(X; Y) = \sum_{i=1}^{n} \sum_{j=1}^{m} p(a_i b_j) log \frac{p(a_i/b_j)}{p(a_i)}$ is labeled as a series.

**Table 1.** The 'LaTeX formulas of advanced mathematics' labeled data.

| Class | Number | Example |
|---|---|---|
| Derivative | 415 | $\frac{d\phi^i}{dt} = -g^{ij}\frac{\partial\omega}{\partial\phi^j}$ |
| | | $\frac{1}{8\pi}\partial_\omega\sum\omega_{ij}^{(2)} + \frac{1}{8\pi}\partial_\Omega\sum\Omega_{ijkl}^{(3)} = \frac{1}{2}$ |
| Integral | 946 | $\delta(x,y) = \int \frac{d^4k}{(2\pi)^4} e^{k^\mu\nabla_\mu\Sigma(x,y)}$ |
| | | $S = \int x\left(\frac{1}{2}\partial_\mu\phi\cdot\partial_\mu\phi + \frac{m^2}{\beta^2}\sum_{a=0}^r n_a e^{\beta\alpha^{(a)}\cdot\phi}\right)^{(2)}$ |
| Series | 193 | $\sum_{S_{p+1}} \theta_{\sigma(i_0)}\theta_{\sigma(i_1)}\cdots\theta_{\sigma(i_p)} = 0$ |
| | | $\sum_{\mu=1}^m F_\mu\left(v_1',\cdots,v_{m-1}'\right) = 0$ |
| Matrix | 305 | $p^0 = \begin{pmatrix} 0 & -i \\ i & 0 \end{pmatrix}$ |
| | | $\begin{pmatrix} z' \\ \bar{z}' \end{pmatrix} = \begin{pmatrix} u & 0 \\ 0 & \bar{u} \end{pmatrix}\begin{pmatrix} z \\ \bar{z} \end{pmatrix}$ |
| Others | 2834 | $V_{\text{eff}}(r) = -\frac{15}{16}\frac{v^4}{r^7} + O\left(\frac{v^6}{r^{11}}\right)$ |
| | | $f(z,\zeta) = \sum_b(\zeta - k)ln\left(\frac{c_b}{(a_b-c_b\zeta)}\frac{[(a_b-c_b\zeta)z+(b_b-d_b\zeta)]}{(c_bz+d_b)}\right) + c'$ |

**Table 2.** 'LaTeX formulas of advanced mathematics' labeled dataset.

| Train Dataset | Validation Dataset | Test Dataset |
|---|---|---|
| 3658 | 457 | 458 |

### 4.2. Paramenters

The word vector dimension of BiLSTM [19], BiGRU [20], and CNN [21] is 300, training epoch 100, learning rate $1 \times 10^{-4}$, and batch size 64. For BERT-related models, we trained them on 12 layer encoders and 12 attention heads, with 100 epochs in total, a batch size of 64, a learning rate of $2 \times 10^{-5}$, and a word vector dimension of 768.

### 4.3. Experiments and Discussion

#### 4.3.1. UL-BERT Experiments

The experiment results are shown in Table 3. The high recalls of 'Matrix' and 'Series' classes are due to their apparent symbols feature and less use in a composite formula. 'Derivative' and 'Integral' classes are easily confused because they are broadly used and mixed with other symbols. The low precision of 'Derivative' is caused by the misclassification of 'Others' and 'Integral' classes, and the misclassification of 'Integral' class affects its recall.

**Table 3.** Results of UL-BERT.

| Class | Precision/% | Recall/% | F1/% |
|---|---|---|---|
| Derivative | 86.36 | 95.00 | 90.47 |
| Integral | 95.83 | 88.46 | 92.00 |
| Series | 95.83 | 100.00 | 97.87 |
| Matrix | 93.75 | 100.00 | 96.77 |
| Others | 96.04 | 96.74 | 96.39 |

#### 4.3.2. Ablation Experiments

Several related models are chosen for verifying the role of pretraining and pseudo-labeling. BERT, BERT with Task pretrained, BERT with Domain pretrained, and TP-BERT

are the different pretrained BERT models for finetune and further self-training. The results are shown in Table 4. 'A-B' means B is used to obtain pseudo-labels, and A uses the obtained pseudo-labels for further learning. For example, BERT-CNN means a CNN is used to generate pseudo-labels for the training of BERT.

**Table 4.** Results of different pretrained models.

| Model (Pretrain Model) | Precision/% | Recall/% | F1/% |
|---|---|---|---|
| BERT | 92.48 | 92.14 | 92.22 |
| BERT (Task pretrained) | 93.60 | 93.45 | 93.47 |
| BERT (Domain pretrained) | 94.00 | 93.88 | 93.91 |
| TP-BERT | 94.26 | 94.10 | 94.14 |
| BERT-BERT | 94.08 | 93.67 | 93.72 |
| BERT-BERT (Task pretrained) | 94.29 | 94.11 | 94.13 |
| BERT-BERT (Domain pretrained) | 95.02 | 94.76 | 94.81 |
| UL-BERT | 95.11 | 94.98 | 94.98 |

BiLSTM, BiGRU, and CNN with GloVe [22] word vectors are chosen as classification models without pretrained. We use three kinds of models to obtain pseudo-labels: classification models (e.g., CNN and LSTM), generative models (e.g., Transformer), and the Bert based self-training model. The results are shown in Table 5.

**Table 5.** Results of different models.

| Model | Precision/% | Recall/% | F1/% |
|---|---|---|---|
| BiLSTM (GloVe) | 85.70 | 85.59 | 85.40 |
| BiGRU (GloVe) | 87.32 | 87.12 | 86.78 |
| CNN(GloVe) | 88.50 | 88.21 | 87.86 |
| BERT | 92.48 | 92.14 | 92.22 |
| MathBERT | 93.29 | 92.87 | 92.97 |
| TP-BERT | 94.26 | 94.10 | 94.14 |
| BERT-BiGRU | 92.99 | 92.58 | 92.65 |
| BERT-BiLSTM | 93.30 | 92.79 | 92.91 |
| BERT- generative model | 93.60 | 93.23 | 93.28 |
| BERT-CNN | 93.73 | 93.45 | 93.50 |
| BERT-BERT | 94.08 | 93.67 | 93.72 |
| UL-BERT | 95.11 | 94.98 | 94.98 |

In Table 6, we fine-tuned MathBERT,TP-BERT and UL-BERT three times with different random seeds, and record the mean, with standard deviations as subscripts.

**Table 6.** Results of different models, with standard deviations as subscripts.

| Model | Precision/% | Recall/% | F1/% |
|---|---|---|---|
| MathBERT | $93.29_{0.38}$ | $92.87_{0.30}$ | $92.97_{0.30}$ |
| TP-BERT | $94.26_{0.44}$ | $94.10_{0.53}$ | $94.14_{0.50}$ |
| UL-BERT | $95.11_{0.14}$ | $94.98_{0.18}$ | $94.98_{0.16}$ |

In Tables 4 and 5 , precision, recall and F1 scores of UL-BERT are the highest among all models, which illustrates the effectiveness of our approach to domain pretraining and self-training.

- The role of BERT
  Comparing with traditional models, such as BiGRU and CNN with GolVe, the F1 score

of the BERT family pretrained models is 5% higher than that of traditional models [23]. The word representation of BERT is produced by their context [24], which significantly improves the model accuracy especially on the ambiguity of LaTeX command words.

- The role of Two-Stage pretraining
  The F1 score of UL-BERT (94.98) and the fine-tuned model with TP-BERT (94.14) is 1–2% higher than that of BERT (92.22), BERT with Task pretrained (93.47), BERT with Domain pretrained (93.91) and MathBERT (92.97). It is shown that the LaTeX formulas require further pretraining to transfer the word representation distribution from the general domain to the LaTeX formula domain.

- The role of pseudo-labeling
  Compared with the original BERT in Table 5, the score of BERT with semi-supervised learning is improved, which proves the effectiveness of semi-supervised learning. Comparing different models that are used to generate pseudo-labels, the Bert based self-training model has the highest F1 score, which proves the effectiveness of self-training.

- Pretraining with pseudo-labels
  UL-BERT (94.98) is almost 1% higher than fine-tuning with TP-BERT (94.14) on the F1 score. UL-BERT combined with TP-BERT and the pseudo-label is better than the other BERTs with the pseudo-label. Pseudo-labels improve the robustness and generalization of classification models that achieve higher F1 scores than pretrained models without pseudo-labels.

### 4.3.3. Comparison of Pseudo-Labels on Threshold

In the DPP method of pseudo-labels, threshold t is an important parameter determining the pseudo-labels' prediction confidence. The F1 scores of UL-BERT with thresholds t 90% or 99% on pseudo-labels are shown in Table 7. All F1 scores of threshold 99% are higher than that of 90%, which means the pseudo-label with the higher prediction probability contains less noise, which optimizes the model's fine-tuning, and pseudo-labels with high confidence are essential for self-training.

**Table 7.** Thresholds in DPP Method of UL-BERT with different pretrained model (%).

| Threshold $t$ | BERT | BERT (Domain Pretrained) | BERT (Task Pretrained) | TP-BERT |
|---|---|---|---|---|
| 90% | 92.23 | 94.34 | 93.66 | 94.12 |
| 99% | 93.72 | 94.80 | 94.13 | 94.98 |

### 4.3.4. Comparison of Low Resource Labeled Data

Pseudo-labeled data can contribute to the model training of low resource labeled data. In these experiments, 10% (360), 20% (720), 30% (1080), 40% (1440), and 50% (1800) from the original labeled data (3658) are defined as low resource data. F1 scores of fine-tuned models and UL-BERT with different proportions of labeled data are shown in Table 8. The F1 score (92.57) of UL-BERT with 50% of the original dataset is better than that (92.22) of the fine-tuned model with BERT on the whole original dataset. All F1 scores of UL-BERT are higher than those of the corresponding fine-tuned model (TP-BERT) without pseudo-labels, which proves that the pseudo-labeling approach is helpful for model training on low resources. When fine-tuned on only 20% of the labels, we achieve 89.32%, outperforming BiLSTM with 100% labels. Meanwhile, as shown in Figure 4, it is concluded that the lower the resources, the more our method (task use of unlabeled data) benefits from multiple iterations.

**Table 8.** Results for low resource data.

| Proportion of Original Labeled Data | Fine-Tuned Model (BERT) | Fine-Tuned Model (TP-BERT) | UL-BERT |
|---|---|---|---|
| 10% | 81.52 | 83.56 | 85.97 |
| 20% | 88.36 | 89.32 | 90.45 |
| 30% | 89.09 | 90.26 | 90.90 |
| 40% | 90.03 | 91.10 | 91.93 |
| 50% | 91.17 | 92.14 | 92.83 |
| 100% | 92.22 | 94.14 | 94.98 |

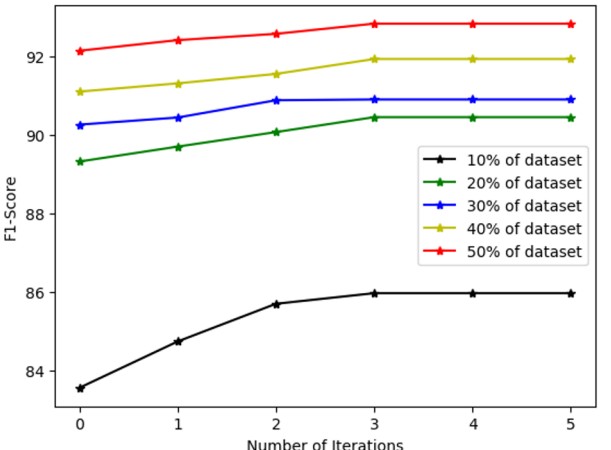

**Figure 4.** Results of iterative training based on T-S under different resource.

### 4.3.5. Results on IMDB

We selected the IMDB dataset [25] in the film review domain to verify the feasibility of our method in other domains. Our framework allows various choices of the Transformer without any constraints. We chose RoBERTa [26] instead of BERT for the IMDB dataset. Roberta, TP-BERT(Roberta), UL-BERT(Roberta) are used to conduct the experiments. The results are shown in Table 9, which prove the feasibility of our method in the film review domain.

**Table 9.** Results on IMDB.

| Model (Pretrain Model) | Precision/% | Recall/% | F1/% |
|---|---|---|---|
| Roberta | 94.13 | 94.11 | 94.11 |
| TP-BERT (Roberta) | 95.20 | 95.19 | 95.19 |
| BERT-BERT(Roberta) | 94.51 | 94.49 | 94.49 |
| UL-BERT(Roberta) | 95.47 | 95.45 | 95.45 |

### 4.3.6. Error Analysis

Table 10 shows some misclassification in our model (UL-BERT). As the formula has a symbol of other categories, the model may misclassify, such as in example 1, with the misclassification of the series as a derivative. The model may pay too much attention to the subscripts of the formulas, which also causes the misclassification, such as in example 1, with the misclassification of the integral as a series. This shows that our model needs to be improved in the face of confusing symbols.

**Table 10.** Examples of some misclassification.

| | Examples | Ground Truth | Prediction |
|---|---|---|---|
| 1 | $(b_r'^{0,1} + \cdots + b_r^{0,n}) \mid \Phi\rangle = 0.$ | Series | Derivative |
| 2 | $E = \frac{1}{n!} \int \left[ E_1 \left| Y_{a_1 \cdots a_n}^{(l)} \right|^2 + E_2 \left| Y_{a_2 \cdots a_n}^{(l)} \right|^2 + E_3 \left| Y_{a_3 \cdots a_n}^{(l)} \right|^2 \right] d\Omega.$ | Integral | Series |

## 5. Conclusions

Obviously, fully utilizing a large amount of unlabeled data is an important and effective way to promote the performance of the classification models of specific domains. The study of the LaTeX formulas demonstrates how to benefit from unlabeled data. The self-training method combined with pretraining is helpful for solving the problem of insufficient resources of labelled data. Experiments show that the proposed UL-BERT can improve the generalization and accuracy of the text classification model.

In our future work, we will conduct more experiments in other domains, such as biomedical and computer science publications, to prove the generalization of our method.In addition, since the cross-entropy loss function can be easily misled by incorrect pseudo-labels, we will use the loss function of contrast learning to modify the loss function of pseudo-labels to mitigate the influence of incorrect pseudo-labels and improve the model performance.

**Author Contributions:** Conceptualization, H.C. and R.Y.; methodology, H.C. and R.Y.; software, R.Y.; validation, H.C. and R.Y.; formal analysis, R.Y.; investigation, R.Y.; resources, H.C. and R.Y.; data curation, R.Y.; writing—original draft preparation, H.C. and R.Y.; writing—review and editing, H.C., Y.T. and T.C.; visualization, R.Y.; supervision, H.C. and Y.F.; project administration, H.C. and Y.F. All authors read and agree to the published version of the manuscript.

**Funding:** This research received no external funding.

**Institutional Review Board Statement:** Not applicable.

**Informed Consent Statement:** Not applicable.

**Data Availability Statement:** Not applicable.

**Conflicts of Interest:** The authors declare no conflict of interest.

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
