# Peer review of "Text Classification Model Enhanced by Unlabeled Data for LaTeX Formula"

_applsci, doi:10.3390/app112210536_

Round 1

Reviewer 1 Report

Dear authors,

this work presents an end-to-end approach that exploits the well-known BERT language model for modeling raw-data documents that contain mathematical formulations and tries to exploit the existence of unlabeled data through an innovative semi-supervised mechanism so as to improve the quality of the finally trained classifier over 5 distinct classes. Such approaches attract the interest of the related community, since the volume of the available data is constantly increasing, settling their manual annotation much more demanding. Thus, the supervision that is provided by automated mechanisms that exploit unlabeled data has been verified as one of the most important tools for tackling this issue. 

However, the work presents some major issues that need clarification, as well as to include some further experiments in order to better understand the importance of the acquired results, which even in its current phase does not seem enough satisfactory, and provide more fair comparisons. I mention them here, while i highlight that all of them need to be tackled for being considered suitable for publication:

  1. The characterization "domain-specific" both in title and into the rest of the manuscript does not seem proper, since this work is clearly focused on mathematical formulations and does not examine any other domain. This should change in the next draft, revealing the current domain.
  2. The first 2 Sections are too short. More information should be provided regarding the examined domain and the spent efforts that could be reduced through an accurate classifier that does not depend on the existence of much labeled instances.
  3. Another great issue is that some really important related work is left untouched. MathBert is of course one of these (https://arxiv.org/abs/2105.00377), while some recently published review works or recent SSL approaches must be described in this Section (https://arxiv.org/abs/2006.05278). The teacher-student strategy should also be introduced here, regarding its recent appearance in the literature and its provided assets.
  4. The notation and the description of the proposed algorithm into Section 3 presents some difficulties and is not smoothly inserted. Please extend your ideas and phrases, and of course improve Figure 3, since it is not obvious how all these stages are combined in the right order.
  5. Next, you must provide your implementation and the specific train-test split of the described dataset, or more correctly, run your experiments 3, 5 or 10 times with different random seeds and provide exactly these batches so as to facilitate the comparison of your approach with competitors, as well as record the mean and std values per measured performance metric. There is no other way for getting confidence regarding your results.
  6. You have also to provide some details regarding the performance of the proposed algorithm and the variants that you mention in the ablation study: how many instances are labeled per iteration for each confidence threshold? How much of them are correctly labeled? 
  7. Mention also some examples that your approach does not classify correctly (error analysis).
  8. Finally, you have to include some SSL variants that exploit other base learners, so as to capture better the difficulty of the task. Since you obtain a feature description through the BERT-based model, you can apply well-known approach of SSL with decision trees, ensemble learners, generative models etc. Look here for some of them:  10.1007/s10115-013-0706-y 10.1142/S0218213017500014, 10.3233/JIFS-152641, 10.1007/s13042-015-0328-7 Moreover, the self-training scheme has recently included also in sklearn library, thus its application to your data should not be much time-consuming. MathBert should also be included into your comparisons.

Reviewer 2 Report

This this the review of the paper titled "Domain-specific Text Classification Model Enhanced by Unlabeled Data" 

This paper presents a good concept of dealing with lack of labeled data. However, still some issues have to be addressed 

1- It is not clear what is the design of the BERT model , how it is trained , components of that model, how do you feed LaTeX formulas to the model , is it as image, vector? if it is vector , how did you convert LaTeX formulas to vector? please explain 

2-  Have you thought about this concept of this paper
https://www.mdpi.com/2072-6694/13/7/1590
You may try it in your paper and you don't labels of pre-trained model 

3- in Figure 3, it is confusing what is the beginning of the flowchart, please give number to the steps 

4- if possible, test your model on a different dataset to show the robustness of the model 
5-  elaborate more on the novelty of the paper? show that there is research gap not address before then the proposed technique solved the issue 

6- comparison with state-of-the-art methods in terms of results is necessary to add. 
7-What are the benefits of this work and in which applications can help. please elaborate on that

Round 2

Reviewer 1 Report

Dear authors,

your work has been improved after the first round of revision. However, some issues still exist and should be tackled for improving the current status.

i) your answer to my 1st issue is not adequate. The fact that your method is also applied to the IMDB dataset does not prove its generalization, while this choice is somehow strange since your work was until then oriented to tackle mathematical-based sources. I heavily recommend taking my previous comment into consideration and change both the abstract and the title so as to clarify your main concern here.

ii) The second issue has been tackled, but the answer to the report is misplaced there. Please be more careful and try to answer the posed questions/advice more precisely.

iii) You have misused in several points into both the manuscript and your review report the use of -ing. This fact has increased dramatically the grammatical errors of this work. Please read carefully the manuscript and elaborate these errors. Also, some issues have been raised through the review

iv) Figure 3 is still weak and confuses the reader about its operation.

v) What does the first column of Table 4 and Table 5 show? Please insert a column-title and avoid arbitrarily numbering or naming, since this put obstacles towards the comprehension of your work.

vi) I also asked to record the mean and std after repeating your experiments more than one time. If you actually did it, you have to record the std, otherwise this statement seems strange, and this kind of recording your results not much professional.

vii) "For different confidence thresholds, 7316 pseudo-labels are generated in each iteration of the Teacher-Student method. Since these pseudo-labels are derived from unlabeled data, we cannot know exactly how many of them are correct pseudo-labels" -> What does this mean? Your collected unlabeled data are probably already labeled but artificially you hide their label. Not any description is provided regarding this answer. In any case, you have to mention how many instances are labeled per iteration and per category, even if you do not have this kind of information, so as to highlight the behavior of all these methods during the learning iterations. 

ix) Consider also the publications that were provided, and of course, expand your last Section mentioning some future plans.

Reviewer 2 Report

Comments were very well addressed 

Author Response

Thanks very much for your kind work and consideration on publication of our paper. On behalf of my co-authors, we would like to express our great appreciation to editor and reviewers.

Round 3

Reviewer 1 Report

Dear authors, 

my main concerns have been resolved. However, there is still space for improvements (i mention some of them), but the current manuscript is adequate for publication.

i) Enhance the Abstract Section adding one sentence for the generic domain, and another one that highlights tour contributions.

ii) 'where generates pseudo-labels' -> where it assigns pseudo-labels

iii) In Table 6, We finetune -> please correct this

iv) remove the numbering from the 1st column of the tables that contain results, the description of the model name is informative enough.

v) 4.3.5. title contains both upper/lower case letters in contrast with the previous subsections' titles.

vi) connect better the experiment on IMDB dataset with the rest manuscript

vii) enhance the last Section 

viii) Rakhlin, A. Convolutional neural networks for sentence classification.GitHub, 2016 this reference should be replaced with its published version: https://aclanthology.org/D14-1181/
